# Mechanisms Involved in the Relationship between Vitamin D and Insulin Resistance: Impact on Clinical Practice

**DOI:** 10.3390/nu13103491

**Published:** 2021-10-01

**Authors:** Victoria Contreras-Bolívar, Beatriz García-Fontana, Cristina García-Fontana, Manuel Muñoz-Torres

**Affiliations:** 1Endocrinology and Nutrition Unit, University Hospital Clínico San Cecilio, 18016 Granada, Spain; victoriacontreras_87@hotmail.com (V.C.-B.); mmt@mamuto.es (M.M.-T.); 2Instituto de Investigación Biosanitaria de Granada (Ibs. Granada), Avd. Madrid 15, 18012 Granada, Spain; 3CIBERFES, Instituto de Salud Carlos III, 28029 Madrid, Spain; 4Department of Medicine, University of Granada, 18016 Granada, Spain

**Keywords:** vitamin D, 25-hydroxyvitamin D or calcidiol (25(OH)D), calcitriol (1,25(OH)_2_D), vitamin D receptor (VDR), 25-hydroxyvitamin D-1alpha-hydroxylase (CYP27B1), insulin resistance, homeostasis model assessment of insulin resistance (HOMA-IR) type 2 diabetes, obesity, metabolic syndrome (MS), polycystic ovary syndrome (PCOS)

## Abstract

Recent evidence has revealed anti-inflammatory properties of vitamin D as well as extra-skeletal activity. In this context, vitamin D seems to be involved in infections, autoimmune diseases, cardiometabolic diseases, and cancer development. In recent years, the relationship between vitamin D and insulin resistance has been a topic of growing interest. Low 25-hydroxyvitamin D (25(OH)D) levels appear to be associated with most of the insulin resistance disorders described to date. In fact, vitamin D deficiency may be one of the factors accelerating the development of insulin resistance. Vitamin D deficiency is a common problem in the population and may be associated with the pathogenesis of diseases related to insulin resistance, such as obesity, diabetes, metabolic syndrome (MS) and polycystic ovary syndrome (PCOS). An important question is the identification of 25(OH)D levels capable of generating an effect on insulin resistance, glucose metabolism and to decrease the risk of developing insulin resistance related disorders. The benefits of 25(OH)D supplementation/repletion on bone health are well known, and although there is a biological plausibility linking the status of vitamin D and insulin resistance supported by basic and clinical research findings, well-designed randomized clinical trials as well as basic research are necessary to know the molecular pathways involved in this association.

## 1. Background

Currently, vitamin D insufficiency affects more than half the population of all ages [1]. The role of vitamin D in bone health is well known. In addition, vitamin D may also play a role in extra-skeletal functions. Vitamin D is a fat-soluble prohormone steroid that has endocrine, autocrine, and paracrine functions [2]. Vitamin D acts as a chemical messenger [3,4] and is involved in the regulation of transcription in approximately 3% of the human genome [5]. Most tissues and organs have receptors for vitamin D, and it appears to be involved in many biological functions. In fact, some studies have shown that low 25(OH)D levels are related to other pathological conditions such as autoimmune diseases, hypertension, cardiovascular disease (CVD) [4,6], and cancer [7]. Insulin resistance has also been linked to vitamin D deficiency. Furthermore, insulin resistance has been linked to multiple disorders such as obesity, type 2 diabetes (T2D), and its complications, MS, and PCOS. In this context, all these diseases could potentially be linked with vitamin D deficiency.

Obesity is generally described as a condition of excessive fat accumulation, with abdominal fat being the main risk factor for insulin resistance [8]. Obesity has become a major public health problem worldwide, being the main cause to the development of diseases such as T2D and CVD [9]. The state of insulin-resistant obesity is often associated with a low circulating concentration of 25(OH)D [10]. However, the possible mechanisms underlying this hypovitaminosis D remain to be elucidated. Furthermore, the role of vitamin D supplementation in obesity is also being studied.

The incidence of T2D is increasingly common and alarming worldwide, in part, to higher obesity rates [11]. The World Health Organization (WHO) reported that most of the diabetes cases (90%) constitute T2D with 15 million affected people worldwide. Moreover, this number could double by 2025 [12]. The systemic inflammation, pancreatic β-cells disfunction, and defect in insulin signaling pathway are engaged in insulin resistance and T2D development [11]. A reduction in the level of some metabolic parameters associated with vitamin D supplementation has been reported by several clinical studies [13,14]. However, the level of 25(OH)D required to obtain an improvement in glycemic homeostasis and to decrease the risk of developing T2D is not fully established.

MS is characterized by a combination of some risk factors such as central (intraabdominal) obesity, hypertension, increased triglyceride (TG) serum levels, decreased high-density lipoprotein cholesterol (HDL-C) levels, hyperglycemia, and insulin resistance [15]. The dysfunction and distribution of adipose tissue has also been considered as an important factor, and the abdominal location of excess adipose tissue has been most closely associated with insulin resistance [16]. The prevalence of MS has increased in recent years, which has been attributed besides to the aging of the population, to the increase in obesity rates related to lifestyle changes, such as low physical activity and poor healthy eating habits [17,18]. It has been proposed that low serum 25(OH)D levels are associated with a higher risk of MS and with the different components that define MS. However, this relationship is not fully established. There is also insufficient scientific evidence, as the results of the different studies are discordant, on the effect of vitamin D supplementation in MS.

PCOS affects up to 25% of women throughout the reproductive years, making it the most common endocrine disorder [19]. PCOS is a heterogeneous disorder, related to metabolic abnormalities such as insulin resistance, systemic inflammation, dyslipidemia, and endothelial dysfunction. The Rotterdam workshop consensus [20] established diagnostic criteria for PCOS based on the combination of at least two of the following three clinical features: hyperandrogenism (clinical and/or biochemical), with acne, androgenic alopecia, and hirsutism; chronic oligo-anovulation; and polycystic ovaries on ultrasound [21]. Although insulin resistance was not included in the Rotterdam criteria, it is a recurrent sign in PCOS. In fact, the key role of compensatory hyperinsulinemia in the onset and progression of PCOS is supported by scientific evidence [22]. Approximately 30–40% of normal-weight PCOS patients and up to 80% of PCOS women with upper body obesity (increased waist-hip circumference) have hyperinsulinemia secondary to insulin resistance [23,24]. There are publications suggesting a molecular implication of 25(OH)D deficiency in insulin resistance, dyslipidemia, inflammation, and decreased fertility, i.e., clinical and metabolic phenomena frequently found in PCOS. However, the molecular mechanism relating 25(OH)D levels and the development of PCOS, as well as vitamin D supplementation and PCOS improvement are currently unknown.

The main circulating form of vitamin D used to determine vitamin D status is calcidiol or 25(OH)D [25]. The definition of adequate 25(OH)D concentrations remains controversial, so there is no clear cutoff point for defining optimal levels of vitamin D. Several studies have proposed that 25(OH)D levels above 30 ng/mL would ensure adequate bone health [26]. However, there has been no evidence regarding which levels of 25(OH)D would be optimal to obtain benefits with respect to glucose and energy metabolism and other health targets [27,28].

Based on these premises, the aim of this review is to summarize the recent evidence linking vitamin D and insulin resistance and related disorders: obesity, T2D, MS, and PCOS. We also analyzed different interventional studies, including vitamin D supplementation, to evaluate its influence on these disorders.

## 2. Methods

A comprehensive search of literature published in PubMed through June 2021 was conducted to identify articles on the 25(OH)D levels, vitamin D supplementation, and insulin resistance. Search strategies were based on the search terms: 25-hydroxyvitamin D, vitamin D intake, vitamin D supplementation, insulin resistance, insulin sensitivity, β-cell function, impaired glucose tolerance, T2D, obesity, MS, and PCOS. A selection of articles published in English providing original human research, observational prospective and retrospective studies, randomized controlled trials, reviews and meta-analyses were included. In addition, we considered case series, single-case reports, editorials, research or original articles, letters to the editor, comments (to an article or from the editor), responses (to a comment, letter, or article), corrections, short reports, short communications, perspectives, opinions, and discussions. Priority was given to the largest studies, and to the strongest available evidence and most recent studies.

## 3. Vitamin D and Insulin Resistance Physiology

The term of vitamin D includes ergocalciferol (vitamin D2) and cholecalciferol (vitamin D3). The main vitamin D metabolites according to their hydroxylation patterns, are calcidiol (25(OH)D) and calcitriol (1,25-dihydroxyvitamin D3 or 1,25(OH)_2_D). Vitamin D in humans comes mainly from the skin and, to a lesser extent, from the diet including foods rich in vitamins D2 and D3 or supplements. Serum vitamin D is bound to vitamin D binding protein (DBP), through which it is transported to the liver where is converted to 25(OH)D by 25-hydroxylase. 25(OH)D leads to 1,25(OH)_2_D, the most biologically active form of vitamin D, in the kidneys through the action of the enzyme 25-hydroxyvitamin D-1alpha-hydroxylase (CYP27B1). The presence of CYP27B1, along with the vitamin D receptor (VDR) in several tissues, suggests that vitamin D could have a key function beyond bone metabolism [29,30].

VDR belongs to the nuclear receptor family participating in DNA transcription [31,32]. Although the VDR acts primarily as a nuclear transcription factor, non-genomic actions of vitamin D have been postulated that involve rapid binding of 1,25(OH)_2_D to cytosolic and membrane VDR that activates several second messenger systems [33]. A vitamin D response element region was identified in the promoter of the insulin receptor gene, so that vitamin D may be involved in the transcriptional control of insulin [34].

Retinoid X receptors (RXRs) are ligand-inducible transcription factors belonging to the superfamily of nuclear receptors which, in the presence of their ligand, can form homodimers or heterodimers with other receptors, including VDR regulating important genes involved in energy homeostasis [35,36,37]. Vitamin A metabolite 9-cis retinoic acid acts as a ligand of RXR and it has been found to be a pancreas-specific autacoid expressed by β-cells capable of exert an effect on the control of glucose levels. Some studies have found increased 9-cis retinoic acid levels associated with mouse models of obesity [38,39]. Based on this, the interaction between VDR, RXR and their ligands could play a relevant role in the pathophysiology of insulin resistance.

The level of glutathione has also been shown to play an important role in the regulation of vitamin D levels. Animal and human studies have found that glutathione is essential for the conversion of provided vitamin D into active vitamin D metabolites: 25(OH)D and 1,25(OH)2D and that it positively regulates the bioavailability of 25(OH)D [40,41]. In addition, vitamin D has also been shown to increase glutathione thus contributing to the reduction of the oxidative stress [42]. Based on these findings, insulin resistance could be related to the glutathione deficiency that exists in many diseases, such as obesity and diabetes [43].

The pathophysiology linking vitamin D and insulin resistance and the four themes involved in this review—obesity, T2D, MS, and PCOS—are detailed in Figure 1.

In summary, vitamin D may play an important role in the regulation and of pancreatic β-cells function in T2D patients [13,14] since calcitriol (1,25(OH)_2_D) acts as a chemical messenger by interacting with calcium flux-regulating receptors on the β-cells [44,45]. Moreover, vitamin D is able to reduce hyperactivity of the renin-angiotensin system and to improve the function of β-cells [46]. On the other hand, vitamin D could influence the insulin secretion regulated by the opening and closing of calcium channels and 1,25(OH)_2_D may also improve insulin sensitivity by stimulating the expression of insulin receptors and activating peroxisome proliferator-activated receptor delta (PPAR-δ) [46]. Finally, the effects of chronic inflammation may be reduced by vitamin D, because vitamin D was shown to deactivate inflammatory cytokines associated with insulin resistance and to promote calbindin expression, leading to protection from apoptosis [47,48].

In the case of obesity, VDR mRNA expression has been identified in visceral and subcutaneous adipose tissue as well as in primary adipocytes. This, coupled with the fact that adipose tissue constitutes a reservoir of vitamin D, suggests that vitamin D plays a key fat-associated role [8,10].

Regarding to MS, vitamin D deficiency may reduce the ability of β-cells to convert proinsulin to insulin [49]. Furthermore, the two most widely accepted hypotheses linking vitamin D and MS are the possible sequestration of vitamin D and its volumetric dilution.

In PCOS, the resulting hyperinsulinemia inhibits hepatic synthesis of sex hormone binding globulin (SHBG), leading to increased circulating free androgens [50,51]. A positive correlation has been found between serum 25(OH)D and SHBG levels. Moreover, an association between vitamin D deficiency with VDR gene polymorphisms has been reported in PCOS. In a review, the authors found that vitamin D was a predictor of insulin resistance in both PCOS and control women [52]. The authors point out that only one study demonstrates no effect of vitamin D3 supplementation on insulin resistance [52,53].

The pathophysiology of vitamin D and insulin resistance in obesity, T2D, MS, and PCOS is detailed.

### 3.1. Vitamin D and Obesity

#### 3.1.1. Obesity and Vitamin D Physiology

VDR is expressed in several cell types, such as pancreatic β-Langerhans cells, liver, and muscle cells, primary pre-adipocytes, and differentiated adipocytes [54]. Thus, 1,25(OH)_2_D plays a central role in multiple metabolic pathways by binding to the VDR [54]. Specifically, VDR mRNA expression has been identified in visceral and subcutaneous adipose tissue of obese and lean individuals showing a higher VDR expression in visceral adipose tissue, in obese than in lean people, whereas no difference in VDR expression have been observed in subcutaneous adipose tissue [55]. In addition, VDR protein expression has recently been demonstrated in primary adipocytes derived from obese donors, indicating a possible role for vitamin D in adipose tissue development and metabolism. In fact, in vitro studies suggest that human adipose tissue growth and remodeling is regulated by calcitriol. Recent studies suggest a role for vitamin D in the adipogenesis, lipogenesis, lipolysis, and inflammation [54,56]. Additionally, vitamin D is a fat-soluble vitamin, being the adipose tissue is a storage site for vitamin D [57]. Thus, overweight and obese subjects show a higher percentage of vitamin D deposited in fat tissue, resulting in reduced serum concentrations of vitamin D in this population compared to lean subjects. Likewise, lower 25(OH)D concentrations have been associated with higher body mass index (BMI). These findings highlight the close relationship between vitamin D and adipose tissue, suggesting that obesity is linked to this [58].

#### 3.1.2. Obesity and Vitamin D Status

Obesity and insulin resistance are tightly interrelated. Inverse correlations have been found between vitamin D status and obesity indicators. In the study of Gonzalez-Molero et al. they found an association between 25(OH)D values ≤ 17 ng/mL and the risk of developing obesity in the following 4 years in non-obese subjects [59]. In the CASPIAN study, a national cross-sectional study, 25(OH)D status was assessed in 2594 students aged 7–18 years [60] which were classified into four metabolic phenotypes of obesity according to BMI and metabolic status categories: “metabolically healthy obese”, “metabolically unhealthy non-obese”, “metabolically unhealthy obese”, and “metabolically healthy non-obese”. The results pointed to hypovitaminosis D as a risk factor for being metabolically healthy obese (OR: 1.46; 95% CI: 1.07–1.77) and metabolically unhealthy obese (OR: 2.89; 1.05–8.31) compared to the healthy group.

Recent ex vivo data published by Di Nisio et al. demonstrated that the adrenaline-mediated lipolytic response, a feature of the obese insulin-resistant state, was accompanied by increased 25(OH)D accumulation in human subcutaneous adipocytes from obese donors, possibly indicating a lower serum 25(OH)D release, suggesting that there may be a link between lipid disturbances and attenuated vitamin D release in obese insulin-resistant individuals [61]. In clinical studies, fat mass reduction has been considered as a strategy to increase circulating 25(OH)D levels in obesity. By contrast, one study found that bariatric surgery by gastric bypass only resulted in a temporary increase in serum 25(OH)D levels, which decreased again in the month following the intervention [62]. Furthermore, some systematic reviews and meta-analyses conclude that weight loss interventions have shown only marginal changes in serum 25(OH)D levels after significant weight and body fat loss [63,64].

Most cross-sectional studies have shown a positive association between serum 25(OH)D levels and measures of insulin sensitivity derived from a hyperinsulinemia euglycemic clamp test or indirect markers such as the quantitative insulin sensitivity index [65]. A negative association has also been observed with HOMA-IR or the 2-h glucose tolerance test (2 h OGTT) in obesity [66,67]. On the other hand, in the study by Muscogiuri et al. the possible relationship between 25(OH)D concentration and insulin sensitivity was analyzed in non-diabetic obese subjects using the glucose clamp technique [68]. The results obtained suggested that there is no cause–effect relationship between vitamin D and insulin sensitivity. These results are also supported by other studies [69,70]. These studies suggest that in obesity, both low 25(OH)D concentration and insulin resistance may be dependent on increased body size. However, a recent review reported that obesity and insulin resistance are often associated with circulating vitamin D deficiency, which may be partly explained by increased sequestration/dilution in adipose tissue [71]. In addition, vitamin D metabolism in adipose tissue may also be altered, as reflected by the downregulation of vitamin D metabolizing enzymes and the finding of an altered release of vitamin D.

#### 3.1.3. Vitamin D and Obesity: Intervention Studies

It has been observed that in obesity there is a deficit of circulating vitamin D, which seems to be associated with a decrease in its bioavailability. It is therefore expected that increasing 25(OH)D levels through supplementation will act to improve insulin sensitivity and function.

Lotfi-Dizaji et al. conducted a study in which vitamin D-deficient subjects (25(OH)D < 20 ng/mL) supplemented with 50,000 IU of cholecalciferol for 12 weeks showed a decrease in weight and fat mass [72]. In a clinical trial involving 200 healthy men with serum 25(OH)D levels < 30 ng/mL, it was found a significant effect of the supplementation treatment (20,000 IU vitamin D3/week/12 weeks) on fasting glucose/insulin ratio compared to the placebo group [73]. The subgroup analyses showed a significant improvement on waist circumference, waist-to-hip ratio, total body fat, and android fat in the group with baseline 25(OH)D levels < 20 ng/mL. In a clinical trial published in 2018, cholecalciferol supplementation (25,000 IU/week vs. placebo for 3 months), combined with a weight loss program, was found to significantly improve insulin sensitivity in metabolically healthy obese subjects, concluding that this therapeutic option could represent a personalized approach for obese subjects with insulin resistance [74]. By contrast, the study by Duan et al. found no beneficial effect of vitamin D supplementation on parameters such as BMI, hip circumference or waist-to-hip ratio [75]. In a randomized, double-blind clinical trial in obese Caucasian adolescents (BMI > 95th), the effect of vitamin D3 supplementation on insulin action and β-cell function was assessed [76]. Subjects were randomly assigned to receive 400 IU/day or 2000 IU/day cholecalciferol. Results showed no effect of vitamin D supplementation, regardless of dose, on β-cell function or insulin action in obese non-diabetic adolescents with relatively good vitamin D status. However, it is unclear whether obese adolescents with vitamin D deficiency and impaired glucose metabolism would respond differently to vitamin D supplementation.

Similarly, a placebo-controlled clinical trial, including 65 overweight or obese adults with vitamin D deficiency (25(OH)D concentration < 20 ng/mL) randomly divided into supplemented (oral bolus dose of 100,000 IU cholecalciferol followed by 4000 IU cholecalciferol/day) and placebo groups for 16 weeks, found no improvements in insulin resistance or secretion parameters associated with vitamin D supplementation [77]. In this line, a meta-analysis including 22 observational studies found that vitamin D supplementation was not found to significantly decrease percentage fat mass relative to placebo, despite the inverse relationship between 25(OH)D levels and percentage fat mass [78]. However, the combined action of vitamin D supplementation (cholecalciferol: 50,000 IU/subject/week) with calcium (1200 mg/calcium/subject/day) seems to increase weight loss and improve some blood metabolic profiles in obese women [79].

A recent review reported that obesity and insulin resistance are often associated with circulating vitamin D deficiency, which may be partly explained by increased sequestration/dilution in adipose tissue [71]. In addition, vitamin D metabolism in adipose tissue may also be altered, as reflected by the downregulation of vitamin D metabolizing enzymes and the finding of an altered release of vitamin D. However, the review by Pramono A et al. indicates that current published meta-analyses do not support vitamin D supplementation for improving glycemic control and insulin sensitivity in overweight/obese subjects [71]. They suggest that ethnic/geographic location and genetic variations in vitamin D-related metabolism may influence the responses and benefits of vitamin D supplementation. In the study of Jamka et al., twelve randomized controlled trials were reviewed, including 1181 individuals with BMI > 23 kg/m^2^ [80]. Vitamin D supplementation (up to 12,000 IU/day) had no effect on glucose concentrations, insulin level, and HOMA-IR values when supplementation dose, time of administration, and baseline 25(OH)D concentration were considered in the sub-analysis. On the other hand, Mirhosseini et al. have recently published a meta-analysis evaluating glycemic outcomes among adults at risk of T2D, including pre-diabetes, overweight, or obesity [81]. Compared to the control group, cholecalciferol supplementation significantly reduced HbA1c level, fasting plasma glucose level, and HOMA-IR level. Subgroup analysis revealed that the effects of vitamin D supplementation on the different glycemic measures were influenced by age, calcium co-administration, vitamin D deficiency, and serum 25(OH)D levels after supplementation and duration of supplementation. Given the results evaluated, they conclude that vitamin D supplementation and improvement of vitamin D status improved glycemic measures and insulin sensitivity and could be useful as part of a preventive strategy for T2D. Reyes-García et al., conducted a prospective study whose objective was to analyze the response of serum 25(OH)D and its predictive factors in healthy postmenopausal women after a dietary intervention with a milk fortified with vitamin D and calcium (900 mg/500 mL) and vitamin D3 (600 IU/500 mL) daily for 24 months [82]. They concluded that baseline 25(OH)D levels were one of the variables that most influenced response to supplementation in addition to fat percentage. In the study of Gröber et al., it is recommended vitamin D supplementation with daily doses of 400–2000 IU of vitamin D to maintain blood levels of 25(OH)D > 30 ng/mL according to the Endocrine Society guidelines in view of the common occurrence of vitamin D insufficiency [83]. The dose of vitamin D in obese adults would be 2–3 times greater to assure blood levels of 25(OH)D > 30 ng/mL due to vitamin D absorption from large body fat stores.

Briefly, it seems clear that where vitamin D deficiency exists, supplementation would be of benefit to overweight and obese subjects. However, further studies are needed to clarify the role of vitamin D supplementation in people who are not vitamin D deficient.

### 3.2. Vitamin D and Type 2 Diabetes

#### 3.2.1. Glucose Homeostasis and Vitamin D Physiology

The role of vitamin D in the maintenance of pancreatic β-cell function has been reported by some in vitro and in vivo studies. This effect could be induced by the activation of VDR located in pancreatic cells. In mice lacking VDR, insulin secretion was impaired, [84] showing a stimulation of the pancreatic islets which results in increased insulin secretion associated with the addition of calcitriol to the culture medium [85].

Moreover, insulin secretion regulated by the opening and closing of calcium channels may be influenced by vitamin D. Calcitriol acts as a chemical messenger by interacting with calcium flux-regulating receptors on the β-cells. Therefore, decreased vitamin D levels may alter normal insulin secretion through calcium flux disturbances [44,45], mainly due to altered flow and calcium concentration across the cell membranes of insulin-responsive tissues [86]. Dephosphorylation of glucose transporter 4 (GLUT-4) can be affected by regulation of extracellular and intracellular calcium concentrations, reducing insulin-stimulated glucose transport [44]. On the other hand, there is evidence that vitamin D can reduce hyperactivity of the renin-angiotensin system and improve the functioning of β-cells [46]. Therefore, insulin resistance pathways associated with diabetes can be improved with adequate vitamin D levels.

1,25(OH)_2_D can improve insulin sensitivity by the stimulation of the insulin receptors expression and by activating PPAR-δ that regulates fatty acid metabolism in adipose tissue and skeletal muscle. One study found that calcitriol could exert a specific effect on the lipid synthesis in the liver and on the glucose production reducing insulin resistance [46].

On the other hand, it has been reported that the effects of chronic inflammation related to the pathogenesis of T2D may be reduced by vitamin D. 1,25(OH)_2_D may have a protection function against cytokine-induced β-cell apoptosis by regulating cytokine expression and activity, improving insulin sensitivity, and may also reduce the effects of inflammation [47]. In addition, vitamin D plays a deactivating role of inflammatory cytokines associated with insulin resistance and promotes the expression of calbindin, thus preventing cell apoptosis [48]. Finally, experimental studies have described the capacity of vitamin D to reduce the accumulation of advanced glycation products which have been linked to insulin resistance and the development of T2D complications [87].

#### 3.2.2. Vitamin D Status and T2D

Serum 25(OH)D concentrations have been found to be inversely related to β-cell function, insulin resistance and glucose homeostasis, and predict a lower risk of both MS and T2D [88,89]. An association between vitamin D insufficiency and the development of insulin resistance has been found in both children [90,91] and adults [92,93]. Overall, the association between vitamin D deficiency and increased incidence of T2D is strongly supported by data from observational studies [94,95]. Indeed, longitudinal studies have found that higher baseline 25(OH)D levels predict lower glucose levels and better β-cell function in subjects at risk of T2D [96].

An Australian study including middle-aged individuals described an inverse association between vitamin D status and the risk of T2D that, apparently, cannot be explained by reverse causality [97]. A 29-year prospective cohort study including 9841 participants, it was found higher risk for T2D development (OR: 1.5 95% CI 1.33–1.70) in the subjects with 25(OH)D levels in the lowest quartile (<5 ng/mL) compared to those in the highest quartile (≥20 ng/mL) [95]. The study of Avila-Rubio et al. reported better glycaemia indices measured by the HOMA in postmenopausal women with 25(OH)D values > 45 ng/dL without established disturbances in glucose metabolism compared to those below [98]. These data are consistent with those of Park et al. [99], showing an inverse dose–response association between 25(OH)D level and the diabetes risk. The authors propose a target 25(OH)D of 50 ng/mL; higher than concentrations suggested in other studies, to improve the incidence rate of diabetes [99].

A cross-sectional study investigating the sex dependence of the association between insulin resistance and serum 25(OH)D levels in a Caucasian population has recently been published [100]. They found that 25(OH)D was inversely and independently associated with insulin resistance only in vitamin D-deficient women, concluding that vitamin D-deficient women may benefit from vitamin D supplementation by improving insulin resistance.

The review on the association between vitamin D deficiency and insulin resistance by Szymczak-Pajor et al. determined that both genomic and non-genomic molecular actions of vitamin D are involved in the maintenance of insulin sensitivity. These favorable effects are not only directly related to insulin signaling, but also indirectly to the reduction of oxidative stress, sub-inflammation, and epigenetic regulation of gene expression and epigenetic regulation of the renin-angiotensin-aldosterone system. Based on the analyzed findings, they concluded that the molecular background of insulin resistance formation is related to vitamin D deficiency [101]. The results revealed that vitamin D deficiency may be a crucial factor that may accelerate the formation of insulin resistance. However, other studies have not confirmed these desired effects of vitamin D on insulin-sensitive tissues. In a study conducted in China with a large sample size, the results supported that there does not appear to be a causal association between vitamin D and pre-diabetes and T2D using a two-way Mendelian randomization approach [102]. Santos et al. conducted a systematic review of the literature on vitamin D status and glycemic control observing an inverse relationship was observed between serum 25(OH)D and indices of glucose metabolism. However, although better glycemic control was related to higher vitamin D levels in T2D patients, the reviewed studies did not show any clear glycemic benefit of vitamin D [103].

A meta-analysis involving 21 observational studies with more than 75,000 subjects revealed 38% lower risk of developing T2D in the reference category with higher 25(OH)D levels compared to the lowest (95% CI 0.54–0.70) [94]. Likewise, the meta-analysis by Mohammadi et al. found an inverse association between serum vitamin D level and the risk of T2D and prediabetes in adults, in a dose–response manner [104]. However, the association was not remarkable for prediabetes. The recently published meta-analysis by Rafiq and Jeppesen concludes that hypovitaminosis D is associated with increased levels of insulin resistance in both T2D patients and healthy people [105].

Moreover, numerous studies have also linked vitamin D deficiency to the occurrence of chronic complications associated with T2D such as macrovascular and microvascular complications and overall mortality [106,107], suggesting that the maintenance of adequate vitamin D status may reduce the risk of mortality in people with diabetes. However, the nature of these studies does not allow us to draw causal conclusions, so we can only speculate about the different associations.

#### 3.2.3. Vitamin D Supplementation and T2D

One point of debate is what levels of 25(OH)D are required to generate an effect on glycemic homeostasis and to reduce the risk of developing T2D. Currently, there is no universal consensus on optimal serum 25(OH)D concentrations or those considered inappropriate. For bone health, there seems to be a relative agreement in considering appropriate values between 20 and 50 ng/mL, deficient values between 12 and 19 ng/mL, and severely deficient values below 12 ng/mL [108]. However, the potential benefits of higher values to achieve extra-osseous benefits is a question currently under investigation [109]. Avila-Rubio et al. suggested that the stated goal of achieving a 25(OH)D level > 30 ng/mL was insufficient to improve glucose metabolism in women with postmenopausal osteoporosis, proposing an optimal level of 45 ng/mL to achieve effects on these targets [98]. Von Horst et al. reported an optimal 25(OH)D concentrations around 50 ng/mL for reducing insulin resistance in Asian women [110]. These data are consistent with a 12-year cohort study in a non-diabetic population in which achieving 25(OH)D values > 50 ng/mL contributed to maximal benefits in reducing the risk of incident diabetes [99]. Similarly, Muñoz-Garach et al. postulate that reaching 25(OH)D levels of 50 ng/mL could improve glucose and insulin homeostasis indices in non-diabetic subjects [111]. It is therefore important to establish what 25(OH)D values are necessary to achieve and, more importantly, maintain the full potential benefits of vitamin. Most studies in this field suggest that the levels established as optimal (30 ng/mL) would not be sufficient to achieve benefits in the prevention of T2D or in the improvement of glucose homeostasis, being necessary levels between 45–50 ng/mL that do not exceed the established toxicity limit. Based on these assumptions, intervention studies are needed to clarify what levels of 25(OH)D need to be achieved.

In recent years, many randomized trials have evaluated the effect of vitamin D supplementation on glucose homeostasis in subjects at risk of T2D showing inconsistent results. Davidson et al. conducted a study in a cohort of African Americans and Latino subjects with hypovitaminosis D and prediabetes [112] which were supplemented with cholecalciferol at a dose sufficient to raise serum 25(OH)D levels to the upper normal range in one group and compared it to a group given placebo. Subjects receiving supplementation reached a mean serum 25(OH)D level of almost 70 ng/mL at 3 months (which was maintained), and there was no change in those receiving placebo. They found no effect on insulin secretion or sensitivity in T2D subjects nor in those with normal glucose tolerance. Sollid et al. compared the supplementation with vitamin D (20,000 IU cholecalciferol weekly) with placebo for one year in subjects with prediabetes for the prevention of T2D [113]. Mean baseline serum 25(OH)D was 23.96 ng/mL and 24.44 ng/mL in the vitamin D and placebo groups, respectively, and increased by 18.32 ng/mL and 1.36 ng/mL, respectively. No significant differences were observed between supplemented and placebo groups in the glycemic or inflammatory markers nor in blood pressure, regardless of baseline serum 25(OH)D concentrations. Forouhi et al. conducted a randomized clinical trial, involving 340 subjects with pre-diabetes or at risk of developing T2D, in which they compared the effect of the supplementation with cholecalciferol or ergocalciferol (both 100,000 IU/month) vs. placebo during four months [114]. The percentages of individuals with a 25(OH)D concentration < 20 ng/mL at baseline were 58.8, 50.9, and 50.9% in the placebo, D2, and D3 groups, respectively; at follow-up these percentages were 47.3, 4.5, and 3.5%. However, there was no difference in HbA1c levels among groups. Thus, it is important to note that only half of the subjects showed circulating 25(OH)D levels < 20 ng/mL, which could influence the results. In the Pittas et al. study, which included people at high risk of T2D not selected for vitamin D insufficiency, vitamin D3 (cholecalciferol) supplementation was given at a dose of 4000 IU/day vs. placebo. Mean baseline 25(OH)D levels were 28.0 ng/mL, with no significant difference between the two groups; 78.3% of participants had a level equal to or greater than 20 ng/mL. The mean 25(OH)D levels in the vitamin D group at month 12 (52.3 ng/mL) and month 24 (54.3 ng/mL) were higher than those in the placebo group (28.1 ng/mL and 28.8 ng/mL). However, despite the increase in 25(OH)D levels, there was no significantly lower risk of diabetes in the supplemented group [115]. In the pre-specified secondary analysis of the DAYLIGHT (Vitamin D Therapy in Individuals at High Risk of Hypertension) randomized controlled trial, the circulating HOMA-IR, high-sensitivity C-reactive protein, pro-B-type N-terminal natriuretic peptide, renin, aldosterone, and lipids at baseline and 6 months were measured in individuals with low vitamin D status (25(OH)D ≤ 25 ng/mL) who received low-dose (400 IU/day) versus high-dose (4000 IU/day) vitamin D3 for 6 months [116]. They found that vitamin D supplementation did not improve biomarkers of blood glucose, inflammation, neurohormonal activation or lipids. Lerchbaum et al. stated that vitamin D supplementation may have a negative effect on insulin sensitivity in healthy men [73]. The study population included 200 healthy men with 25(OH)D levels < 30 ng/mL receiving 20,000 IU vitamin D3 or placebo at week for 12 weeks. A significant effect of the supplementation was found only on fasting glucose/insulin ratio. The study by Wallace et al., a randomized, double-blind, placebo-controlled trial, aimed to investigate the effect of vitamin D3 supplementation (3000 IU daily for 26 weeks) on insulin resistance and β-cell function in people with pre-diabetes and suboptimal vitamin D status (<20 ng/mL) [117]. Baseline serum 25(OH)D concentrations in the vitamin D3 and placebo group were 12.28 and 12 ng/mL, with status increasing by 28.2 ng/mL and 2.12 ng/mL, respectively, after supplementation. The results indicated that vitamin D supplementation had no effect on insulin action in people with pre-diabetes. The meta-analysis by Tang et al. found no effect of vitamin D supplementation on the T2D incidence [118]. However, the authors proposed a possible dose–response effect of vitamin D supplementation postulating a possible benefit of higher vitamin D doses for primary prevention of T2D. It should be noted that in most studies, vitamin D supplementation has not reached 25(OH)D levels of 45–50 ng/mL, so probably no beneficial results beyond the maintenance of adequate bone health have been obtained. This would support that higher levels of 25(OH)D may be necessary to obtain extraosseous effects aimed at improving glucidic homeostasis.

In this context, Talari et al. conducted a study in vitamin D-deficient diabetic patients with ischemic heart disease and found that a schedule of 50,000 IU of vitamin D every 2 weeks for 6 months in combination with omega-3 fatty acids resulted in a significant reduction of fasting blood glucose levels and increased insulin sensitivity [119]. Mitri et al. found a significant improvement in insulin secretion in overweight or obese prediabetic subjects supplemented with cholecalciferol 2000 IU daily and calcium carbonate for four months compared to the placebo group [120]. In this line, Gagnon et al. reported an improvement in insulin sensitivity indices after supplementation with cholecalciferol 2000–6000 IU daily and calcium carbonate 1200 mg only in subjects with prediabetes but not in those with glucose intolerance or newly diagnosed diabetes [121]. Barzegari et al. conducted a parallel, randomized, double-blind, placebo controlled clinical trial in 50 subjects with diabetic nephropathy [122], of which half of them received treatment with 1,25-dihydroxycholecalciferol (50,000 IU/week) for 8 weeks. They found significantly increased vitamin D levels associated with vitamin D supplementation observing significant decreased TG, LDL-C and TC serum levels in the supplemented group compared to placebo with no changes in oxidative/antioxidative markers and HDL-C levels. On the other hand, diabetes contributes to atherosclerosis partly by inducing oxidative stress. VDR and RXR receptor agonists are known to have antiatherogenic effects. In a study using a mouse model of diabetes, the effects of combined treatment with VDR and RXR agonists on the progression of atherosclerosis and the mechanisms involved were evaluated [123]. The supplementation with calcitriol (200 ng/kg, twice/week) and bexarotene (10 mg/kg, daily) alone or in combination for 12 weeks showed a delay in the progression of atherosclerosis independently of serum lipid and glucose levels. Additionally, the co-administration of VDR ligand (1,25(OH)_2_D) and the RXR ligand 9-cis retinoic acid produced synergistic protection against glucose-induced endothelial cell apoptosis concluding that the preventive effects of the RXR agonist may be partially dependent on VDR activation.

Angelotti and Pittas reviewed the role of vitamin D in the prevention of TD2 [124], concluding that there is a strong and consistent inverse association between blood 25(OH)D concentration and incident diabetes reported in observational studies and supported by data on the biological plausibility of mechanistic studies. They suggest that vitamin D supplementation may have a role in the prevention of T2D in high-risk populations. However, they point that since T2D is a multifactorial disease it is unlikely that vitamin D deficiency is the main cause of the development of T2D. Therefore, the hypothesis that vitamin D contributes to the pathogenesis of T2D and has a role in prevention remains to be tested, suggesting that future studies are needed to support its use as a therapeutic target in T2D.

The meta-analysis conducted by Barbarawi et al. found that in patients with pre-diabetes, vitamin D supplementation at moderate or high doses (≥1000 IU/day) significantly reduced the risk of T2D incidence compared to placebo [125]. Agreeing, *Muñoz-Garach* et al. proposed that vitamin D supplementation at doses close to 4000 IU/day could be a suitable strategy to improve glucose and insulin homeostasis indices in non-diabetic subjects. However, they note that there is no consensus as to whether the general population needs additional vitamin D supplementation to improve health outcomes. Therefore, the establishment of the specific populations that could benefit from nutritional recommendations regarding the calcium and vitamin D intake has become a field of special interest at the moment [111].

Regarding vitamin D supplementation in pregnant women with gestational diabetes, a recent meta-analysis suggests that vitamin D supplementation may lead to improved glycemic control and reduced adverse maternal–neonatal outcomes [126].

Although most randomized clinical trials did not show a beneficial effect of vitamin D supplements on glycemic homeostasis, insulin sensitivity indices, and T2D incidence and its complications in subjects at risk of diabetes [7,77,112,115,121,127,128], there is some interesting evidence to support a beneficial effect of vitamin D on β-cell function [119,125].

The different studies were heterogeneous in terms of duration and type of supplements, study population characteristics, and design. Adherence to treatment is likely to play an important role in the interpretation of the results. Thus, it is important to note that the 25(OH)D level to be achieved may be higher in the T2D population and, therefore, many studies have not found beneficial results on glucidic metabolism. Furthermore, it seems that the studies that have found an effect on glucidic metabolism have used higher doses of vitamin D supplementation.

On the other hand, some studies has proposed that vitamin D supplementation in combination with l-cysteine may be more effective than vitamin D alone in reducing the risk of oxidative stress and inflammation associated with T2D obtaining better results for the treatment of insulin resistance [129]. In this context, studies in animal models evaluating the involvement of L-cysteine in the treatment of T2D have reported that L-cysteine supplementation may provide a novel approach to increase blood levels of DBP and 25(OH)D in TD2 [130]. Consistently, results from several studies have shown that vitamin D in combination with L-cysteine may be more successful in increasing glutathione and vitamin D metabolism genes constituting an effective strategy for the treatment of vitamin D deficiency and insulin resistance compared to vitamin D supplementation alone [41,42,129,131].

Therefore, combined vitamin D and L-cysteine supplementation may be a promising approach to maximize the guarantee of success in intervention studies treating insulin resistance.

Briefly, there are numerous factors that appear to influence the effect of vitamin D supplementation on insulin resistance, such as the plasma 25(OH)D levels to be achieved, the types and doses of vitamin D to be administered, or the administration of vitamin D alone or in combination with other components (calcium, L-cysteine, among others).

### 3.3. Vitamin D and Metabolic Syndrome

#### 3.3.1. Metabolic Syndrome and Vitamin D Physiology

Several pathophysiological mechanisms have been proposed to understand the possible relationship between vitamin D and MS. However, the pathophysiological mechanism linking MS to vitamin D remains unknown. One plausible explanation is that vitamin D influences insulin secretion and sensitivity, which plays an important role in the development of MS. Vitamin D deficiency may compromise the ability of β cells to convert proinsulin to insulin [49]. Another pathophysiological mechanism that could be related is the association between obesity and vitamin D deficiency. Although obesity is not a defining criterion for MS, increased abdominal waist is. Increased abdominal waist is usually present in overweight subjects. Therefore, another pathophysiological mechanism is the association between obesity and vitamin D deficiency, as obesity is present in most people with MS. MS is closely related not only to obesity, but also to T2D, so everything discussed above about vitamin D and T2D and obesity applies to MS. The two most widely accepted hypotheses linking vitamin D and MS are vitamin D sequestration and volumetric dilution [57]. This is because vitamin D is a fat-soluble vitamin, having an affinity for adipose tissue. Thus, in obese subjects, where there is an increase in adipose tissue, vitamin D deficiency may be due to vitamin D sequestration by adipose tissue. Other possible explanations for this relationship could be based on poor lifestyle habits, such as decreased sun exposure, inadequate diet, differential gene expression of vitamin D metabolizing enzymes, or impaired hepatic 25-hydroxylation [132,133].

#### 3.3.2. Vitamin D Status and Metabolic Syndrome

25(OH)D levels have been associated with the different components of MS. A study in non-diabetic young people showed an inverse association between the presence of MS and 25(OH)D levels, due to the combined effect of obesity and insulin resistance [134]. Lee et al. showed an increased MS risk with low levels of 25(OH)D in both Korean men and women older than 65 years [135]. After adjusting for area of residence, season, exercise, smoking, alcohol, and age, there appeared to be a relationship between low 25(OH)D levels and a higher prevalence of MS, such that the lower the 25(OH)D levels, the higher the prevalence of increased waist circumference, hypertriglyceridemia, and increased LDL-C levels. Similarly, decreased 25(OH)D levels has been related to the prevalence and incidence of MS in Spanish population [136], showing lower mean 25(OH)D levels in subjects with MS than in those without (21.7 ng/mL vs. 23.4 ng/mL). Moreover, lower 25(OH)D levels were found to be related to individual components of the MS observing more frequent hypertension hyperglycemia, increased waist circumference and hypertriglyceridemia in men and women. However, in the follow-up period of the patients for 5 years, vitamin D deficiency was not significantly associated with an increased risk of developing MS after adjusting for sex and age. In this line, Mehri et al. indicated that the absence of long follow-up did not allow a causal relationship between inadequate 25(OH)D levels and T2D to be established with certainty [137]. Barbalho et al. observed hipovitaminosis D in 80% of patients at the cardiology unit, all of them having MS [138]. In addition, they also found significantly higher BMI, blood glucose levels, glycosylated hemoglobin, TG, TC, LDL-C, and atherogenic indices in patients with vitamin D deficiency compared to those with adequate levels of vitamin D.

However, some researchers question the relationship between vitamin D deficiency and the components defining MS. A recent study found discordant associations between 25(OH)D levels and biochemical and genetic parameters related to the risk of developing T2D [139].

The available literature on retrospective studies suggests that there seems to be an association between 25(OH)D deficiency and the different components of MS, however, more longitudinal studies are needed to support this association. Given the cross-sectional data, it seems reasonable to determine serum 25(OH)D in at-risk subjects to detect whether they are vitamin D deficient. If deficiency is present, restoration of 25(OH)D levels could be attempted, and an assessment of whether beneficial effects are observed.

#### 3.3.3. Vitamin D and Metabolic Syndrome: Intervention Studies

To date, the literature presents contradictory results on the effects of vitamin D supplementation in MS patients, similar to occurs with T2D and pre-diabetes as previously discussed. Regarding the relationship between vitamin D levels with the different components of MS, the available literature points to a possible beneficial effect of vitamin D supplementation in overweight or obese subjects with hypovitaminosis D, as described in the section on obesity.

Regarding high blood pressure, some studies have reported a significant reduction in blood pressure associated to vitamin D administration [140,141]. In the work of Golzarand et al., the dose of vitamin D3 supplementation could be 800 IU/day or >800 IU/day, the duration of the intervention also varied between 6 months or >6 months, and the treatment regimens could be daily or intermittent; in addition the supplementation could have calcium added or not [140]. They found that the results indicated that a daily dose of vitamin D3 > 800 IU for <6 months can significantly reduce systolic and diastolic blood pressure. In addition, vitamin D3 was found to have a significant hypotensive effect in healthy subjects and in patients with hypertension. In work by Rajakumar et al., this beneficial effect was found with doses of 1000 IU or 2000 IU of vitamin D3 vs. 600 IU per day for 6 months [141]. However, other analyses conducted in randomized clinical trials reported no significant effect on systolic or diastolic blood pressure values [142,143].

On the other hand, vitamin D supplementation may also improve the lipid profile as reported by Jamilian et al., who found a reduction of TG and very low-density lipoprotein cholesterol (LDL-C) associated with 50,000 IU vitamin D supplementation every 2 weeks for 6 months [144]. Imga et al. also found an improvement in HOMA-IR and LDL-C in both obese and overweight women supplemented with vitamin D3 for 6 months observing [145]. The beneficial effects of vitamin D on HDL-C levels have been supported by the meta-analysis performed by Ostadmohammadi et al. [146]. Although many studies suggest that vitamin D has a beneficial effect on the lipid profile, there are also studies in the opposite direction. Thus, Farrokhian et al. found no significant changes in lipid profile after vitamin D supplementation, although they reported changes in plasma malonaldehyde levels, which results from lipid peroxidation [147]. Similarly, AlAnouti et al. reports inconsistent results on the relationship between vitamin D status and dyslipidemia in MS adults, pointing mainly to a lack of effect, despite improved vitamin D status [148]. However, the authors indicate that these results should be interpreted with caution given the limited number of included clinical trials RCTs, the small sample size, and the limited intervention period.

Considering the available scientific evidence, the review by Melguizo-Rodriguez et al. concluded that vitamin D deficiency appears to be associated with the different components that define MS [149] suggesting that vitamin D supplementation could be an appropriate strategy in the treatment of MS. However, although there seems to be an association between vitamin D and MS, it is not possible to reach a solid conclusion on this association, as there is still controversy among some of the studies in this field, and it is not clear whether vitamin D deficiency is a cause or an effect of MS or any of its components.

Several authors agreed on the need for vitamin D supplementation to maintain adequate 25(OH)D levels to reduce the risk of MS and associated diseases [150,151]. To date, the level, form, or dose of 25(OH)D needed to reach a beneficial effect on this goal is not known. Various daily or monthly cholecalciferol dosing regimens have led to adequate results. Carbonare et al. found that 80% of patients receiving 1750 IU/day or 50,000 IU/month for six months achieved 25(OH)D levels > 30 ng/mL [152]. However, it may be necessary to achieve 25(OH)D levels greater than 30 ng/mL to achieve beneficial effects in MS, in addition to those related to bone health. It is therefore of great importance to establish the optimal level of vitamin D needed to prevent the risk of MS. Finally, it seems essential to determine 25(OH)D levels in at-risk subjects and to be able to develop supplementation interventions, as well as to implement public health programs on healthy habits to prevent vitamin D deficiency.

### 3.4. Vitamin D and Polycystic Ovary Syndrome

#### 3.4.1. Polycystic Ovary Syndrome and Vitamin D Physiology

Despite extensive research into the etiology of PCOS, it currently remains largely unknown. Insulin resistance and hyperandrogenism, which are consequences of insulin resistance, have been proposed as key factors in the pathogenesis of PCOS [153,154]. The resulting hyperinsulinemia inhibits hepatic synthesis of SHBG, leading to an increased circulating free androgens [50]. On the other hand, insulin resistance and hyperinsulinemia contribute to elevated circulating androgens through direct stimulatory effects on ovarian theca and by increasing ovarian androgen synthesis [155]. Hyperandrogenemia feeds back to worsen insulin resistance, generating a vicious cycle that perpetuates itself. In addition, high insulin levels are also involved in central adiposity, a phenomenon that is more prevalent in PCOS women compared to control women [156]. Moreover, in recent years, anti-Mullerian hormone (AMH), an important marker of ovarian reserve, has been considered as a possible cause of PCOS. Serum levels of AMH have been found to be significantly higher in women with PCOS and have been correlated with circulating androgen levels, as well as insulin resistance [157].

Some observational and experimental studies provide compelling evidence linking vitamin D deficiency to many of the endocrine, metabolic, and clinical components of PCOS. Vitamin D deficiency has been associated with the most prevalent phenomena in PCOS, such as hyperandrogenism [158], insulin resistance [101], adiposity indices [159], systemic proinflammatory indices, and ovulatory dysfunction [160,161]. Moreover, vitamin D is a steroid hormone with progesterone-like activity [162]. However, the mechanisms underlying the association between vitamin D and PCOS are not fully understood.

Few studies have evaluated VDR polymorphisms and/or polymorphisms related to vitamin D metabolism in women with PCOS in relation to insulin resistance [161], vitamin D status, and metabolic disturbances. An association between vitamin D deficiency with VDR gene polymorphisms in PCOS and its endocrine-metabolic alterations has been suggested [163]. A genetic variation in the VDR may affect PCOS development, as well as insulin resistance in women with PCOS [164].

On the other hand, several circumstances related to of insulin resistance, including PCOS, are associated with lower levels of SHBG [50,51]. SHBG is a transporter protein that regulates free androgen levels [165]. A positive correlation has been found between serum 25(OH)D and SHBG levels [166]. In addition, data from in vitro assays with human adrenocortical cells provide convincing evidence for the suppressive effect of vitamin D on steroidogenic cells, with a consequent decrease in the levels of steroid intermediates [167].

Recently, Krul-Poel et al., in a review, found that univariate regression analyses of weighted means revealed that vitamin D was a predictor of insulin resistance in both PCOS and control women [52]. However, significance disappeared after adjusting for BMI in women with PCOS. In this review, the authors also point out that only one study demonstrates no effect of vitamin D3 supplementation on insulin resistance [53].

Although some studies point to a link between vitamin D deficiency and the components of PCOS, mainly due to insulin resistance, it is not yet clear whether it is the main cause of PCOS or whether these are independent features in women with PCOS.

#### 3.4.2. Vitamin D Status and Polycystic Ovary Syndrome

About 67–85% of women with PCOS show a vitamin D deficiency [168]. Vitamin D deficiency appears to be implicated in several features of PCOS, such as insulin resistance, hirsutism, infertility, and cardiovascular risk [169,170].

Women with PCOS and vitamin D deficiency have a higher prevalence of glucose intolerance compared to women with PCOS without vitamin D deficiency [168]. In addition, as noted above, vitamin D deficiency also appears to have an impact on insulin sensitivity in women with PCOS, as assessed by HOMA-IR [163,169,170]; however, further assessment of insulin sensitivity by hyperinsulinemia euglycemic clamp in women with PCOS did not confirm this association [171]. In addition to insulin resistance, vitamin D deficiency has been associated with cardiovascular risk factors, such as higher systolic and diastolic blood pressure, increased C-reactive protein, higher total cholesterol and TG, and lower HDL-C [169]. Women with PCOS and hirsutism have been found to have lower 25(OH)D levels than BMI-matched controls [169,172], which may be explained by an association of vitamin D with androgens or SHBG [170,173]. In a cross-sectional study in adolescents, participants with PCOS had higher levels of AMH and lower levels of 25(OH)D than in the group without PCOS [174]. They concluded that since traditional clinical markers of PCOS may be physiological in adolescents, AMH and 25(OH)D can be used as surrogate markers of PCOS risk in this population. Wehr et al. reported that women with normal ovulation had higher vitamin D levels than women with PCOS [169]. In fact, 25(OH)D deficiency has been associated with lower rates of follicle development and gestation following stimulation in women with PCOS [175]. Thus, vitamin D supplementation may improve reproductive function in women with PCOS, by acting to restore normal menstrual cycles [163]. Butts et al. conducted a retrospective cohort study that aimed to assess the relationship between vitamin D deficiency and reproductive outcomes after ovarian stimulation in women with PCOS or unexplained infertility [176]. To test their hypothesis, they performed an assessment of vitamin D status in stored sera from randomized controlled trials conducted by the Reproductive Medicine Network: the Pregnancy in Polycystic Ovary Syndrome II (PPCOS II) trial [177] and the Assessment of Multiple Intrauterine Gestations by Ovarian Stimulation (AMIGOS) trial [178]. Finally, they observed that in PPCOS II, subjects with vitamin D deficiency (25(OH)D < 20 ng/mL) were less likely to ovulate and experienced a 40% lower chance of having a live birth than those without deficiency. However, in subjects from the AMIGOS trial, no significant association was observed between 25(OH)D deficiency and live births. The study of Lumme et al., which included 1246 women, showed that women with PCOS were no more prone to vitamin D deficiency than non-symptomatic controls, since a considerable number of women in both groups had low 25(OH)D levels with mean 25(OH)D concentrations above the normal range in participants with self-reported PCOS [179]. Nevertheless, the authors of this paper recommend that sufficient vitamin D levels should be ensured in women with PCOS, especially in overweight and obese women.

The review by Muscogiuri et al. concluded that there may be a relationship between deficient vitamin D and PCOS and its components [180]. However, they indicate that further studies, adjusting for confounding factors, are needed to clarify this possible relationship. In the recent review by Di Bari et al. they indicate that there is an association between low 25(OH)D levels and obesity, hyperandrogenism, insulin resistance, and other metabolic dysfunctions related to PCOS [181]. Furthermore, they report that bone health may be influenced by various facets of PCOS, which may result in an increased risk of fracture over time and further exacerbated by hypovitaminosis D, as it is directly and indirectly related to poor bone health in PCOS. In the meta-analysis of Bacopoulou et al., which included 2262 women (1150 PCOS patients and 1162 controls), they found that serum 25(OH)D, follicle-stimulating hormone and SHBG were significantly lower in PCOS patients than in controls. In addition, they also observed that the HOMA-IR, serum insulin, TC, LDL-C, TG, LH, and testosterone were significantly higher in PCOS patients compared to controls.

Based on the available evidence, an inverse association between vitamin D status and metabolic and hormonal alterations in PCOS has been reported. However, given the variability of PCOS phenotypes and the heterogeneity of available studies, it is difficult to draw clear conclusions. Further studies are needed to clarify the relationship between vitamin D deficiency and PCOS and, if confirmed, what levels of 25(OH)D would be appropriate for benefit.

#### 3.4.3. Vitamin D and Polycystic Ovary Syndrome: Intervention Studies

The molecular mechanism between vitamin D supplementation and PCOS improvement is currently unknown. Recent studies have found positive effects of vitamin D supplementation in PCOS. Indeed, it has been suggested that vitamin D supplementation may attenuate the deleterious effects of advanced glycation end products (AGEs) in PCOS by increasing androgen synthesis and enhancing abnormal folliculogenesis. This could be due to the role of vitamin D in attenuating the adverse effects of AGEs on steroidogenesis by granulosa cells, possibly by down-regulating the expression of the cell membrane receptor for AGEs [182,183].

Kadoura S. et al. conducted a randomized, placebo-controlled clinical trial in 40 women with PCOS with 25(OH)D deficiency (<30 ng/mL) [184]. Participants were assigned to take metformin (1500 mg/day) plus placebo, or metformin (1500 mg/day) plus calcium (1000 mg/day) and vitamin D3 (6000 IU/day) orally for 8 weeks. They investigated the effects of combining supplements (vitamin D and calcium) with metformin on menstrual cycle abnormalities, gonadotrophins, and the IGF-1 system. They observed that calcium and vitamin D supplementation appear to support the effect of metformin in regulating menstrual cycle irregularity in PCOS patients with 25(OH)D deficiency or insufficiency, however this effect was not associated with significant changes in gonadotrophins or the IGF-1 system. Irani et al. found that vitamin D supplementation resulted in a significant decrease in abnormally elevated AMH levels in vitamin D-deficient women with PCOS [185]. A double-blind, randomized, placebo-controlled trial, including 180 women with PCOS to receive vitamin D (20,000 IU/week) or placebo for 24 weeks [186], showed a decrease in plasma glucose one hour after the oral glucose tolerance test (OGTT) in the supplemented group although no significant effect on metabolic and endocrine parameters (menstrual frequency, testosterone, TC, TG, HbA1c, HOMA-IR, nor insulin sensitivity according to -QUICKI-) was found.

In the recent review by Iervolino et al., they conclude that vitamin D appears to be effective in the treatment of PCOS [187]. Accordingly, the review by Kalyanaraman and Pal concluded that based on the available evidence, vitamin D supplementation has a recognized safety profile and could therefore be considered a safe and cost-effective intervention to mitigate biochemical and clinical stigma, as well as the ovarian hyperstimulation syndrome in PCOS [188].

Regarding glucose homeostasis, the systematic review of Łagowska et al. concluded that there is evidence that co-supplementation with doses below 4000 IU/day vitamin D (cholecalciferol) in women with PCOS is effective in lowering fasting glucose concentration [189] and in decreasing HOMA-IR. Consistently, a recent meta-analysis by Guo et al. included thirteen clinical trials (824 patients) aimed at finding out the effect of vitamin D supplementation on various metabolic parameters in PCOS [190]. The authors reported that vitamin D supplementation alone seems to be associated with a significant reduction in fasting plasma glucose. However, heterogeneity among studies was high, mainly due to the types of vitamin D supplements. The beneficial effect on plasma glucose astringency was found to be greater when considering daily intake compared to weekly independently of baseline vitamin D deficiency among the patients. Moreover, vitamin D supplementation seems to be associated to the improvement of insulin resistance, and of lipid profile, observing a significant reduction in serum VLDL-C levels in supplemented patients compared to placebo.

A systematic review assessing the effect of vitamin D supplementation on circulating AMH in women with PCOS [191] revealed a complex cause–effect relationship associated with the ovulatory status. Vitamin D supplementation was associated with a decrease in AMH levels in patients with an-ovulatory PCOS but with increased AMH levels in the ovulatory PCOS population.

In the meta-analysis by Miao et al., the effect of vitamin D supplementation on BMI, total testosterone, dehydroepiandrosterone sulphate (DHEAs), TG, TC, or lipoprotein-cholesterol, HOMA-IR and the model of cell function (HOMA-B) was assessed in 483 women [192]. Available data on vitamin D supplementation suggest that it may reduce insulin resistance and hyperandrogenism in PCOS patients. However, the results did not show a positive effect of vitamin D supplementation on BMI, TG levels, HDL-C, or DHEAs.

Concluding, the available literature appears to indicate a positive effect of vitamin D supplementation in PCOS patients. However, further studies are needed to draw conclusive results about the role of vitamin D in the pathogenesis of PCOS, as most studies have not adjusted for confounding factors that may affect vitamin D status, such as dietary intake, intake of additional nutrients co-integrated with vitamin D, or factors that determine vitamin D deficiency (sun exposure, physical exercise, etc.). Until then, it seems reasonable to screen women with PCOS at risk for vitamin D deficiency and prescribe vitamin D supplementation if they are deficient in vitamin D. Supplementation seems likely to improve different aspects of PCOS such as BMI, insulin resistance, lipid profile, cardiovascular risk, menstrual regularity, fertility, and bone health. Given these benefits, its low cost, and safety, vitamin D supplementation could be considered as one of the therapeutic options for women with PCOS, in addition to insulin-sensitizing agents and antioxidants, regardless of BMI.

## 4. Unsolved Questions and Conclusions

Diseases associated with insulin resistance are becoming increasingly common. Recent findings suggest that the molecular background to the development of insulin resistance may be related to vitamin D deficiency. Taken together, the results of basic and clinical studies reveal that vitamin D deficiency may be a key factor triggering the insulin resistance. In this review, a large body of findings on vitamin D and its association with disorders related to insulin resistance such as obesity, T2D, MS, and PCOS has been analyzed, with controversial results. So far, numerous observational studies and randomized trials involving very heterogeneous populations have been conducted, differing in design, duration, and in the types and doses of vitamin D. In addition, there are factors, such as glutathione deficiency, which could play a role in the action of vitamin D on insulin resistance. This review highlights the need to clarify the level of vitamin D required to obtain a tangible benefit, if any. This concentration is probably higher than the current recommendations focused mainly on achieving bone metabolism benefits. Although currently there is no consensus as to whether vitamin D supplementation is needed in the general population to improve health outcomes, vitamin D supplementation at doses approaching 600–4000 IU/day could be an option to increase 25(OH)D levels close to 50 ng/mL to improve insulin resistance and the associated disorders.

Establishing whether specific populations such as those with obesity, MS, prediabetes, T2D, and/or PCOS could benefit significantly from nutritional recommendations regarding vitamin D intake has become a matter of particular interest. However, what does seem to be clear is the need to determine the level of 25(OH)D in high-risk subjects and to supplement in case of deficiency, which will undoubtedly bring a benefit and provide more data to draw more solid conclusions.

## Figures and Tables

**Figure 1 nutrients-13-03491-f001:**
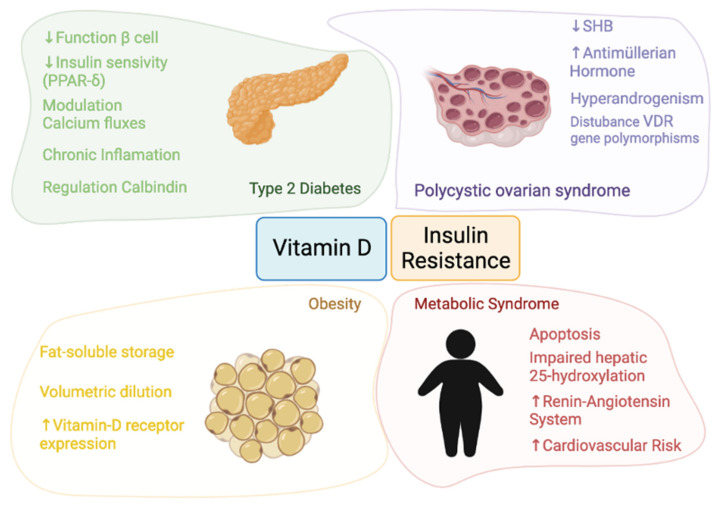
Pathophysiology of the relationship between vitamin D and insulin resistance.

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
