# Peer review of "Mechanisms Involved in the Relationship between Vitamin D and Insulin Resistance: Impact on Clinical Practice"

_nutrients, 2021, doi:10.3390/nu13103491_

Round 1
Reviewer 1 Report
This manuscript summarizes various studies in literature that has examined the intervention of vitamin D supplementation on insulin resistance. Overall, this is a systematic review. This review does conclude that although there is a link between vitamin D deficiency and insulin resistance, well designed clinical trials are needed to understand the molecular pathways associated with this link. Reviewer aggress with the overall discussion and conclusion. Some suggests given below will be helpful to the readers and strengthen the review.
Specific recommendation:
- This is a topic of significant research interest to the public because both vitamin D deficiency and IR is an epidemic in our populations.
- Some discussion on why some vitamin D supplementation clinical studies are not successful is needed. Numerous studies with large patient population reports that vitamin D supplementation clinical trials are not completely successful as has been mentioned by the authors, such as in diabetic patients (PMID: 33052876, 29671531, 32244496, 28490514). Glutathione deficiency exists in many diseases including diabetes and obesity (PMID 19166318). Glutathione is essential for the conversion of consumed vitamin D into active metabolites of vitamin D, 25-hydroxy-vitamin D and 1-25 dihydroxy-vitamin D (PMID 30578920, 31616013, 30160165). Vitamin D needs to be converted to active metabolites for its efficacy. Vitamin D also increases glutathione and reduces oxidative stress (PMID: 23770363). Various studies have shown that vitamin D in combination with L-cysteine can be more successful in increasing glutathione and vitamin D metabolism genes and decreasing insulin resistance therefore is efficacious in treating of vitamin D-deficiency and insulin resistance compared with vitamin D alone (PMID 30160165; 33596158, PMID 26778482; PMID 33171932). Combined supplementation with vitamin D and L-cysteine may be a promising approach in providing successful treatment for IR. These references needs to be discussed and its inclusion in text of discussion will strengthen this paper and help readers and provide potential explanation for why vitamin D supplementation studies are not successful.
Author Response
Reviewer #1:
This manuscript summarizes various studies in literature that has examined the intervention of vitamin D supplementation on insulin resistance. Overall, this is a systematic review. This review does conclude that although there is a link between vitamin D deficiency and insulin resistance, well designed clinical trials are needed to understand the molecular pathways associated with this link. Reviewer aggress with the overall discussion and conclusion. Some suggests given below will be helpful to the readers and strengthen the review.
First, we want to thank for your effort in reviewing our manuscript and for your constructive comments, which have undoubtedly contributed to improve the quality of our manuscript. Please, find below the responses to your kind suggestions:
Specific recommendation:
Comment 1: This is a topic of significant research interest to the public because both vitamin D deficiency and IR is an epidemic in our populations.
We fully agree with your statement. Vitamin D deficiency and insulin resistance and its associated disorders (reviewed in our manuscript) are very frequent, with an increasing prevalence. Therefore, we think it is of great importance to know whether there might be a link between them.
Comment 2: Some discussion on why some vitamin D supplementation clinical studies are not successful is needed. Numerous studies with large patient population reports that vitamin D supplementation clinical trials are not completely successful as has been mentioned by the authors, such as in diabetic patients (PMID: 33052876, 29671531, 32244496, 28490514). Glutathione deficiency exists in many diseases including diabetes and obesity (PMID 19166318). Glutathione is essential for the conversion of consumed vitamin D into active metabolites of vitamin D, 25-hydroxy-vitamin D and 1-25 dihydroxy-vitamin D (PMID 30578920, 31616013, 30160165). Vitamin D needs to be converted to active metabolites for its efficacy. Vitamin D also increases glutathione and reduces oxidative stress (PMID: 23770363). Various studies have shown that vitamin D in combination with L-cysteine can be more successful in increasing glutathione and vitamin D metabolism genes and decreasing insulin resistance therefore is efficacious in treating of vitamin D-deficiency and insulin resistance compared with vitamin D alone (PMID 30160165; 33596158, PMID 26778482; PMID 33171932). Combined supplementation with vitamin D and L-cysteine may be a promising approach in providing successful treatment for IR. These references need to be discussed and its inclusion in text of discussion will strengthen this paper and help readers and provide potential explanation for why vitamin D supplementation studies are not successful.
We would like to thank you for this feedback as it has allowed us to enrich our knowledge of the regulatory mechanisms of vitamin D and to further explore the mechanisms by which vitamin D supplementation is not as successful as initially proposed in some studies.
Considering the new information provided by the reviewer, the manuscript has been modified in the following sections to include it:
- In Section “ Vitamin D and Insulin resistance physiology”, a paragraph describing the relation between vitamin D and glutathione levels (PMID: 33052876, 29671531, 32244496) has been included as follows:
“The level of glutathione has also been shown to play an important role in the regulation of vitamin D levels. Animal and human studies have found that glutathione is essential for the conversion of provided vitamin D into active vitamin D metabolites: 25(OH)D and 1,25(OH)2D and that it positively regulates the bioavailability of 25(OH)D [40][41]. In addition, vitamin D has also been shown to increase glutathione thus contributing to the reduction of the oxidative stress [42]. Based on these findings, insulin resistance could be related to the glutathione deficiency that exists in many diseases, such as obesity and diabetes [43].”
- In subsection "1.3. Vitamin D and Obesity: intervention studies" between lines 269-275, we have included the following paragraph (reference PMID: 28490514):
“Similarly, a placebo-controlled clinical trial including 65 overweight or obese adults with vitamin D deficiency (25(OH)D concentration < 20 ng/mL) randomly divided into supplemented (oral bolus dose of 100,000 IU cholecalciferol followed by 4,000 IU cholecalciferol/day) and placebo groups for 16 weeks found no improvements in insulin resistance or secretion parameters associated to vitamin D supplementation [77].”
- The information related to the following bibliographic references (PMID 19166318, 30578920, 31616013, 30160165, 23770363, 30160165; 33596158, PMID 26778482; PMID 33171932) has been included in the section "2.3. Vitamin D supplementation and T2D" to expand the discussion on the absence of effects of vitamin D supplementation:
“On the other hand, some studies has proposed that vitamin D supplementation in combination with l-cysteine may be more effective than vitamin D alone in reducing the risk of oxidative stress and inflammation associated with T2D obtaining better results for the treatment of insulin resistance [129]. In this context, studies in animal models evaluating the involvement of L-cysteine in the treatment of T2D have reported that L-cysteine supplementation may provide a novel approach to increase blood levels of DBP and 25(OH)D in TD2 [130]. Consistently, results from several studies have shown that vitamin D in combination with L-cysteine may be more successful in increasing glutathione and vitamin D metabolism genes constituting an effective strategy for the treatment of vitamin D deficiency and insulin resistance compared to vitamin D supplementation alone [41][42][129][131].
Therefore, combined vitamin D and L-cysteine supplementation may be a promising approach to maximize the guarantee of success in intervention studies treating insulin resistance.
Briefly, are numerous factors that appear to influence the effect of vitamin D supplementation on insulin resistance, such as the plasma 25OHD levels to be achieved, the types and doses of vitamin D to be administered, or the administration of vitamin D alone or in combination with other components (calcium, L-cysteine, among others).”
- Finally, in the section " Unsolved Questions and conclusions" we have added the following sentence (lines 878-880):
“In addition, there are factors, such as glutathione deficiency, which could play a role in the action of vitamin D on insulin resistance.”
Reviewer 2 Report
The review work presented here entitled “Mechanisms involved in the relationship between vitamin D and insulin resistance: impact on clinical practice” is well written, clear, and easy to read. The topic is consolidated and adds further information to the subject area related to metabolic disorders compared with other published work.
From a biochemistry point of view please indicate 25-hydroxyvitamin D as 25(OH)D not as 25OHD, either for its active metabolite 1,25-dihydroxy vitamin D3 or calcitriol 1,25(OH)2D not as 1,25(OH)2D.
Please adds a subsection concerning the molecular mechanism of VDR which form heterodimers with RXR receptor. In this latter concern, 9cis retinoic acid was found to be an autacoid pancreas-specific so far.
Author Response
Reviewer #2:
The review work presented here entitled “Mechanisms involved in the relationship between vitamin D and insulin resistance: impact on clinical practice” is well written, clear, and easy to read. The topic is consolidated and adds further information to the subject area related to metabolic disorders compared with other published work.
First, we want to thank for your effort in reviewing our manuscript and for your constructive comments, which have undoubtedly contributed to improve the quality of our manuscript.
You can find our response to your main comments below:
Comment 1: From a biochemistry point of view please indicate 25-hydroxyvitamin D as 25(OH)D not as 25OHD, either for its active metabolite 1,25-dihydroxy vitamin D3 or calcitriol 1,25(OH)2D not as 1,25(OH)2D.
In agreement with the reviewer's suggestion, we have modified 25-hydroxyvitamin D as 25(OH)D instead of 25OHD and 1,25-dihydroxyvitamin D3 or calcitriol as 1,25(OH)2D. These changes are marked in orange colour throughout the text.
Comment 2: Please adds a subsection concerning the molecular mechanism of VDR which form heterodimers with RXR receptor. In this latter concern, 9cis retinoic acid was found to be an autacoid pancreas-specific so far.
We are very grateful that you have given us this insight into the pathophysiology of vitamin D and insulin resistance, in which the VDR and RXR receptors play a key role.
This information has been added to the section “3. Vitamin D and Insulin resistance physiology", including the corresponding bibliographical references as follows:
“Retinoid X receptors (RXRs) are ligand-inducible transcription factors belonging to the superfamily of nuclear receptors which, in the presence of their ligand, can form homodimers or heterodimers with other receptors, including VDR regulating important genes involved in energy homeostasis [35–37]. Vitamin A metabolite 9-cis retinoic acid acts as a ligand of RXR and it has been found to be a pancreas-specific autacoid expressed by β-cells capable of exert an effect on the control of glucose levels. Some studies have found increased 9-cis retinoic acid levels associated with mouse models of obesity [38][39]. Based on this, the interaction between VDR, RXR and their ligands could play a relevant role in the pathophysiology of insulin resistance.”
In addition, we have added the following paragraph regarding the study by Lin et al, in relation to the VDR and RXR agonist treatment in the prevention of the atherosclerosis in diabetes in the subsection "3.2.3. Vitamin D supplementation and T2D" as follows:
“On the other hand, diabetes contributes to atherosclerosis partly by inducing oxidative stress. VDR and RXR receptor agonists are known to have antiatherogenic effects. In a study using a mouse model of diabetes, the effects of combined treatment with VDR and RXR agonists on the progression of atherosclerosis and the mechanisms involved were evaluated [123]. The supplementation with calcitriol (200 ng/kg, twice/week) and bexarotene (10 mg/kg, daily) alone or in combination for 12 weeks showed a delay in the progression of atherosclerosis independently of serum lipid and glucose levels. Additionally, the co-administration of VDR ligand (1,25(OH)2D) and the RXR ligand 9-cis retinoic acid produced synergistic protection against glucose-induced endothelial cell apoptosis concluding that the preventive effects of the RXR agonist may be partially dependent on VDR activation.”